# Effects of Linguistic Labels on Learned Visual Representations in Convolutional Neural Networks: Labels matter!

## Abstract

We investigated how the visual representations learned by CNNs is affected by training using different linguistic labels (e.g., basic-level labels only, superordinate-level only, or both at the same time), and how these differently-trained models compare in their ability to predict the behavior of humans tasked with selecting the object that is most different from two others in a triplet. CNNs used identical architectures and inputs, differing only with respect to the labels used to supervise the training. In the absence of labels, we found that models learned very little categorical structure, suggesting that this structure cannot be extracted purely from the visual input. Surprisingly, models trained with superordinate labels (vehicle, tool, etc.) were most predictive of the behavioral similarity judgments. We conclude that the representations used in an odd-one-out task are highly modulated by semantic information, especially at the superordinate level.

## 1 Introduction

A critical distinction between human category learning and machine category learning is that only humans have a language. A language means that human learning is not limited to a one-to-one correspondence between a visual input and a category label. Indeed, the users of a language are known to actively seek out categorical relationships between objects and use these relationships in making perceptual similarity judgments and in controlling behavior (Hays, 2000; Lupyan & Lewis, 2017). A premise of our work is that a language provides a semantic structure to labels, and that this structure contributes to the superior efficiency and flexibility of human vision compared to any artificial systems (Pinto et al., 2010). Of course, the computer vision literature on zero-shot and few-shot learning has also made good progress in leveraging semantic information (e.g., image captions, attribute labels, relational information) to increase the generalizability of a model's performance (Lampert et al., 2013; Sung et al., 2018; Lei Ba et al., 2015).

Still, this performance pales in comparison to the human ability for classification, where zero-shot and few-shot learning is the norm, and efficiently-acquired category knowledge is easily generalized to new exemplars (Ashby & Maddox, 2005; Ashby & Ell, 2001). One reason why machine learning lags behind human performance may be because of a failure to fully consider the semantic structure of the ground-truth labels used for training, which can be heavily biased by basic or subordinate-level categories. This might result in models learning visual feature representations that may not be best for generalization to new, higher-level categories. For example, ImageNet (Deng et al., 2009) contains 120 different dog categories, making the models that are trained using these labels dog experts, creating an interesting but highly atypical semantic structure.

Here we study how the linguistic structure of labels influences what is learned by models trained on the same visual inputs. Specifically, we manipulated the labels used to supervise the training of CNN models, each having the same architecture and given identical visual inputs. For example, some of these models were trained with basic-level labels only, some with only superordinate-level labels, and some with both. We then compare visual representations learned by these models, and predict human similarity judgement that we collected using an Odd-one-out task where people had to select which of three object images was the most different. With this dataset, and using categorical representations extracted from our trained models, we could predict human similarity decisions with

up to 74% accuracy, which gives us some understanding of the labels needed to produce human-like representations. Our study also broadly benefits both computer vision and behavioral science (e.g., psychology, neuroscience) by suggesting that the semantic structure of labels and datasets should be carefully constructed if the goal is to build vision models that learn visual features representations having the potential for human-like generalization. For behavioral science, this research provides a useful computational framework for understanding the effect of training labels on the human learning of category relationships in the context of thousands of naturalistic images of objects.

## 2 RELATED WORK

NEW

### 2.1 SEMANTIC LABEL EMBEDDING

Although many computer vision models perform well in image classification, generalization tasks such as zero-shot and few-show learning remain challenging. Several studies have attempted to address this problem by embedding semantic information into a model's representations using text description (Lei Ba et al., 2015), attribute properties (Lampert et al., 2013; Akata et al., 2015; Chen et al., 2018), and relationships between objects (Sung et al., 2018; Annadani & Biswas, 2018). More related to our work, some studies even directly leveraged the linguistic structure of labels. For example, Lei et al. (2017) and Wang & Cottrell (2015) found that training CNNs with coarse-grained labels (e.g., basic-level categories) improve classification accuracy for finer-grained labels (e.g., subordinate-level labels). Also, Frome et al. (2013) re-trained a CNN to predict the word vectors learned by a word embedding model, instead of using one-hot labels, and found improved zero-shot predictions; the model was able to predict thousands of novel categories that were never seen with 18% accuracy. These results suggest that different semantic structures of labels, such as word hierarchy, an order of learning, or semantic similarity between words, affect learned visual representations in CNNs to differing degrees. The current study provides a more systematic investigation of this question.

### 2.2 UNDERSTANDING HUMAN VISUAL REPRESENTATION

The human visual system is unparalleled in its ability to learn feature representations for objects that are robust to large changes in appearance. This tolerance to variability, not only enables accurate object recognition, but also facilitates generalization to new exemplars and categories(DiCarlo et al., 2012). Understanding how humans learn these visual representations is, therefore, an enormously important question, but one that is difficult to study because human learning in the real world is affected and confounded by many factors that are difficult to control experimentally. Recently, work has addressed this issue by computationally modeling and simulating human representation. For example, Hebart et al. (2019) studied human visual representations by fitting probabilistic models to human similarity judgement, and found that human visual representations are composed of semantically interpretable units, with each conveying categorical membership, functionality, and perceptual attributes. Peterson et al. (2018), the study most similar to ours, trained CNNs with labels that differed in hierarchy (e.g., subordinate-level vs. basic-level). They found that training on coarser-grained labels (either as standalone or as coming after finer-grained) induces a more semantically structured representation, and produces more human-like generalization performance. The current study builds on this earlier work by 1) including CNNs trained with no labels (autoencoder) or very fine-grained labels (word vector), 2) testing on a large-scale dataset of human similarity judgement, and 3) comparing superordinate vs. basic levels.

## 3 MODEL TRAINING

Our goal is to study how linguistic labels change the visual representations learned by CNNs. To do this, we trained equivalently designed CNNs for classification, but each with different linguistic labels as ground-truth. In addition, we trained a Convolutional autoencoder, which encodes the images using the same convolutional structure as the other models but, instead of being supervised to predict the class of the image, the objective of this model is to generate an output image that is the same as the input. This Conv. Autoencoder, therefore, represents a model that was not trained with any linguistic label, in contrast to the other models that were each trained with some type of linguistic labels. The description of each model and the labels used for training are provided below.

- **Conv. Autoencoder**: Autoencoder with Convolutional encoder and decoder trained to output the same image as input
- **Basic labels**: CNN model trained with one-hot encoding of basic-level categories, n=30
- **Superordinate labels**: CNN model trained with one-hot encoding of superordinate-level categories, n=10
- **Basic + Superordinate**: CNN model trained with two-hot encoding of both basic and superordinate-level categories, n=40(10+30)
- **Basic then Superordinate**: CNN model trained with one-hot encoding of basic-level categories first (n=30), and then finetuned with one-hot encoding of superordinate categories (n=10)
- **Superordinate then Basic**: CNN model trained with one-hot encoding of superordinate-level categories first (n=10), and then finetuned with one-hot encoding of basic categories (n=30)                                                                                          NEW
- **Basic FastText vectors**: CNN model trained with basic-level word vectors extracted from FastText word embedding model (Bojanowski et al., 2017), dimension=300
- **Superordinate FastText vectors**: CNN model trained with superordinate-level word vectors extracted from FastText word embedding model (Bojanowski et al., 2017), dimension=300                                                                                                   NEW

The identical CNN architecture was used for each model in our labeling manipulation, except for the output layer and its activation function. This general pipeline is described in Figure 1. Our CNN models consist of five blocks of two Convolutional layers, each followed by Max pooling and Batch normalization layers. For all Convolutional and Max pooling operations, zero padding was used to produce output feature maps having the same size as the input. Rectified linear units (ReLU) were used to obtain an activation function after each convolution. The flattened output of the final Convolutional layer, the "bottleneck" feature that we later extract and use as a model's visual representation (dim=1568), was then fed into one fully connected dense layer. For Conv. Autoencoder, the same Convolutional architecture was used for encoding and decoding, with the hidden layer in the model (dim=1568) serving as the bottleneck feature for analysis. The final predicted output, "label vector" is either one-hot or word embedding according to the model's target labels. Output activation functions differed depending on what label vector was used: a sigmoid function for Basic + Superordinate CNN, a linear function for the Conv. Autoencoder and FastText vectors CNNs, and a softmax for the rest of CNNs.

All models were trained and validated on the images of 30 categories from the IMAGENET 2012 dataset (Deng et al., 2009), and tested on images of the same 30 categories from the THINGS dataset (Hebart et al., 2019). These 30 basic-level categories were grouped into 10 higher-level, superordinate categories, which included: 'mammal', 'bird', 'insect', 'fruit', 'vegetable', 'vehicle', 'container', 'kitchen appliance', 'musical instrument', and 'tool'. A list of all 30 categories, with their superordinates, are provided in the Supplementary 7.1. All input images were converted from RGB to BGR and each channel was zero-centered with respect to the ImageNet images. Different loss functions were used for training different models: Binary Crossentropy loss for Basic + Superordinate CNN, and Mean Squared Error loss for both Conv. Autoencoder and FastText vectors CNNs, and Categorical Crossentropy loss was used for the rest of the CNNs. All models were trained using Adam optimization (Kingma & Ba, 2014), with a mini-batch size of 64. During training, early stopping was implemented and the model with the lowest validation loss was used for the following analysis.

## 4 BEHAVIORAL DATA

To compare the visual representations learned by our trained models with those of humans, we collected human similarity judgments in an Odd-one-out task, as in Zheng et al. (2019). Participants were shown three images of objects per trial, a triplet, and were asked to choose which object was most different from the other two. Each triplet consisted of three exemplar objects from the 30 categories used for our model training. All exemplar objects came from Zheng et al. (2019), except for 'crate', 'hammer', 'harmonica', and 'screwdriver', which were replaced with new exemplars to increase image quality and category representativeness. There are 4060 possible triplets that can be

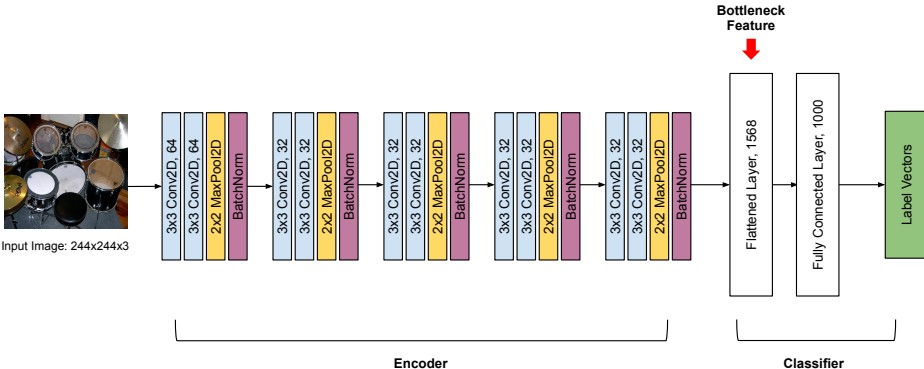

Figure 1: Pipeline for the CNNs used for the study. The bottleneck features (the flattened output of the final Convolutional layer) are later extracted and used as a model's visual representation (dim=1568). The final predicted output, "Label Vector" is either one-hot or word embedding according to the model's target labels.

generated from all 30 categories, but we collected behavioral data on only a subset of these to reduce the time and cost of data collection. This subset includes 1) the ten triplets having objects coming from the same superordinate category, e.g., 'orangutan', 'lion', 'gazelle' 2) all 435 triplets where two objects came from the same superordinate category, e.g., 'orangutan', 'lion', 'minivan', and 3) 1375 triplets where all objects came from different categories, e.g., 'orangutan', 'minivan', 'lemon', yielding 1820 unique triplets in total. 51 Amazon Mechanical Turk (AMT) workers participated in this task, each making responses on ∼200 triplets. After removing responses with reaction times below 500ms, we collected 9697 similarity judgments where each triplet was viewed by 5.6 workers, on average (min=4, max=51).

Table 1: **Classification accuracy for trained models.** Exact match accuracy is the same as top@2 accuracy from Basic + Superordinate CNN and the same as top@1 accuracy for the other models. Detailed architecture of models are described in Section 3 and Figure 1. Average precision and average recall scores are reported in Supplementary 7.3

| Model | # of classes | Output dimension | Accuracy | | |
|---|---|---|---|---|---|
| | | | exact match | top@3 | top@5 |
| Basic labels | 30 | 30 | 0.90 | 0.98 | **0.99** |
| Superordinate labels | 10 | 10 | **0.95** | **0.99** | **0.99** |
| Basic + Superordinate | 40 | 40 | 0.91 | 0.95 | 0.97 |
| Basic then Superordinate | 10 | 10 | **0.95** | **0.99** | **0.99** |
| Superordinate then Basic | 30 | 30 | 0.88 | 0.97 | **0.99** |
| Basic FastText vectors | 30 | 300 | 0.52 | 0.74 | 0.82 |
| Superordinate FastText vectors | 10 | 300 | 0.77 | 0.94 | 0.96 |

## 5 EXPERIMENTS

### 5.1 EVALUATING MODEL PERFORMANCE

Although our goal was not to compete with state-of-the-art vision models in classification, we evaluated classification accuracy to see the effects of different labels on learning, thereby confirming that the visual features learned by our models represented category knowledge. To evaluate classification accuracy, we report top@k, the percentage of accurately classified test images where the

true class was in the model's the top K predictions in Table 1. Average precision and average recall over all categories are also reported in the Supplementary 7.3. All metrics were computed on the THINGS test dataset (Hebart et al., 2019). Because the FastText vectors CNN predicts a word vector, not a class, we approximated its classification performance by calculating cosine similarity between predicted and true word vectors and choosing the corresponding class from top@k similarities. Classification results cannot be generated from Conv. Autoencoder, but we include examples of images generated from this model in the Supplementary 7.2 to show that the model worked. As can be seen in Table 1, the top@5 classification accuracy for all trained models was good (all $>.82$), although there is room for improved classification for FastText vectors CNN.

## 5.2 EXPLORING VISUAL REPRESENTATIONS

To explore how the different linguistic labeling schemes affected the learned visual representations, we extracted and analyzed the bottleneck features from each model (i.e., the 1568-dimensional output of the last Convolutional layer; see Figure 1). We first measured the representational similarity of all objects in the training dataset (IMAGENET 2012; Deng et al., 2009) both between and within each category. These representational distributions were visualized using t-SNE (Maaten & Hinton, 2008) and are attached in Supplementary 7.5. We also analyzed the similarity between categorical    FIX representations by plotting a similarity matrix in Figure 2. To create categorical representations, we simply averaged the obtained bottleneck features from all training images per category, creating in a sense "prototypical" representation for each class.

### Clustering Quality

To investigate how model's category representations are dense and well separated, we computed the ratio of between-category dispersion and within-category dispersion using cosine distance (1-cosine angle of two feature vectors). Between-category dispersion is the average cosine distance between the center(mean) of different categories. Within-category dispersion is the average cosine distance between every exemplar and the center of each category. Comparing the models in Table 2 revealed that using distributed word vectors as targets, especially Superordinate FastText vectors, produced the highest between-to-within ratio, suggesting the most tightly clustered representations. Interestingly, the Basic + Superordinate CNN model, which was trained with both basic and superordinate labels at the same time, learned more scattered and less distinguishable categorical representations compared to other label-trained models. Lastly, Conv. Autoencoder produced the lowest between-to-within ratio, suggesting that even if a model learns visual features that are good enough to generate input-like images, these visual representations may still be poorly discriminable not only in basic level categories, but also in superordinate level categories. Widely distributed features of Conv. Autoencoder from T-SNE plots in Supplementary 7.5 further supported that the visual input alone is not sufficient to produce any clusterable structure or category representations. A similar trend was observed in the other clustering quality measures as reported in the Supplementary 7.4.

Table 2: **Comparison of clustering quality.** between: between-category dispersion in cosine distance; within: within-category dispersion in cosine distance; ratio: between-to-within dispersion ratio. Larger values indicate model's visual representations having dense and well separated category clusters

| Model | By superordinate category | | | By basic category | | |
|---|---|---|---|---|---|---|
| | between | within | ratio↑ | between | within | ratio↑ |
| Conv. Autoencoder | 0.02 | 0.19 | 0.11 | 0.03 | 0.19 | 0.15 |
| Basic labels | 0.36 | 0.55 | 0.64 | 0.43 | 0.52 | 0.84 |
| Superordinate labels | 0.33 | 0.47 | 0.71 | 0.36 | 0.46 | 0.80 |
| Basic + Superordinate | 0.29 | 0.48 | 0.61 | 0.35 | 0.45 | 0.78 |
| Basic then Superordinate | 0.40 | 0.53 | 0.76 | 0.46 | 0.51 | 0.90 |
| Superordinate then Basic | **0.42** | **0.56** | 0.75 | **0.49** | **0.53** | 0.93 |
| Basic FastText vectors | 0.36 | 0.37 | 0.95 | 0.40 | 0.35 | 1.14 |
| Superordinate FastText vectors | **0.42** | 0.38 | **1.11** | 0.44 | 0.37 | **1.18** |

**Visualization of Categorical Representations**

Figure 2 visualizes cosine similarity matrices for the category representations learned by the models to explore whether the hierarchical semantic structure of the 30 categories is captured (e.g., every basic-level category belongs to one of ten superordinate categories). For a complete comparison, we also analyzed categorical representations extracted from SPoSE (Zheng et al., 2019), FastText (Bojanowski et al., 2017), and VGG16 early layer (i.e., the output from the first max-pooling layer; Simonyan & Zisserman, 2014). SPoSE model's category representations were trained on human similarity judgments. This serves as an approximation of human perceived similarity, which can be a combination of semantic and visual similarities. While FastText similarity represents the semantic similarity between categories in basic-level terms, VGG16 early layer similarity represents lower-level visual similarity. Whereas little effect of category hierarchy can be seen in VGG16 early layer or Conv. Autoencoder features, the various semantic structure can be observed in the other models (e.g., the emergent bright yellow squares in the figure). Upon closer analysis, these categorical divisions seemed to occur for 1) nature vs. non-nature, 2) edible vs non-edible, and 3) the superordinate categories. Surprisingly, basic-level structures are still observed in Figure 2f (e.g., fine-grained lines in the diagonal), where the model is trained only on the superordinate-level labels. This suggests that guidance from superordinate labels was often as good or better as guidance from much finer-grained basic-level labels, which is consistent with the previous finding that training with coarser labels induce more hierarchical structure in visual representations (Peterson et al., 2018)

## 5.3 PREDICTING HUMAN VISUAL BEHAVIOR

Finally, we evaluated how well the visual representations learned by the models could predict human similarity judgement in an Odd-one-out task (See Section 4). For each triplet, responses were generated from the models by comparing the cosine similarities between the three visual object representations and selecting the one most dissimilar from the other two. Three kinds of visual representations were computed and compared: 1) IMAGENET categorical representations, where features were averaged over ∼1000 images per category from the IMAGENET training dataset (Deng et al., 2009), 2) THINGS categorical representations, where features were averaged over ∼10 images per category from the THINGS dataset (Hebart et al., 2019), and 3) Single Exemplar representation, where only one feature per category was generated for the 30 exemplar images used in the behavioral data collection. Together with accuracy from SPoSE (Zheng et al., 2019), FastText (Bojanowski et al., 2017), and VGG16 early layer (Simonyan & Zisserman, 2014), three baseline models of accuracy are reported below, which constitute upper and lower bounds.

- **Null Acc**: Accuracy achieved by predicting that every sample is the most frequent class in the dataset (lower bound, 36%).
- **Bayes Acc**: Accuracy achieved by predicting that every sample is the most frequent class in each unique triplet set (upper bound, 84%).
- **SPoSE Acc**: Accuracy achieved using the SPoSE model (Zheng et al., 2019), a probabilistic model that is directly trained on human responses on all triplets from 1854 THINGS objects (80%).

As shown in Figure 3, triplet prediction accuracy was highest when models used IMAGENET category representations and lowest when single exemplar representations were used, even if exemplar image is the one that participant actually saw during the experiment. This shows that when humans do visual similarity ratings, they not only evaluate visual inputs but also use rich and abstract semantic information learned from viewing myriad exemplars. Comparing individual model performance, the highest accuracy (74%) was obtained by the model trained with superordinate labels. This performance is particularly impressive, considering 1) how coarsely grained superordinate labels are (dim=10) compared to Basic labels (dim=30), Basic + Superordinate labels (dim=40), or FastText vectors (dim=300), and 2) that this model is not trained on the actual human triplet data, as was the case for the SPoSE model whose performance was about 80%.

These results suggest that the representations used by humans in an Odd-one-out task are highly semantic, reflecting category structure, especially at the superordinate level. However, this may be only because the setting of odd-one-out task has caused people to use superordinate label information. For example, when the participants are given a triplet like ('orangutan', 'lion', and 'lemon'),

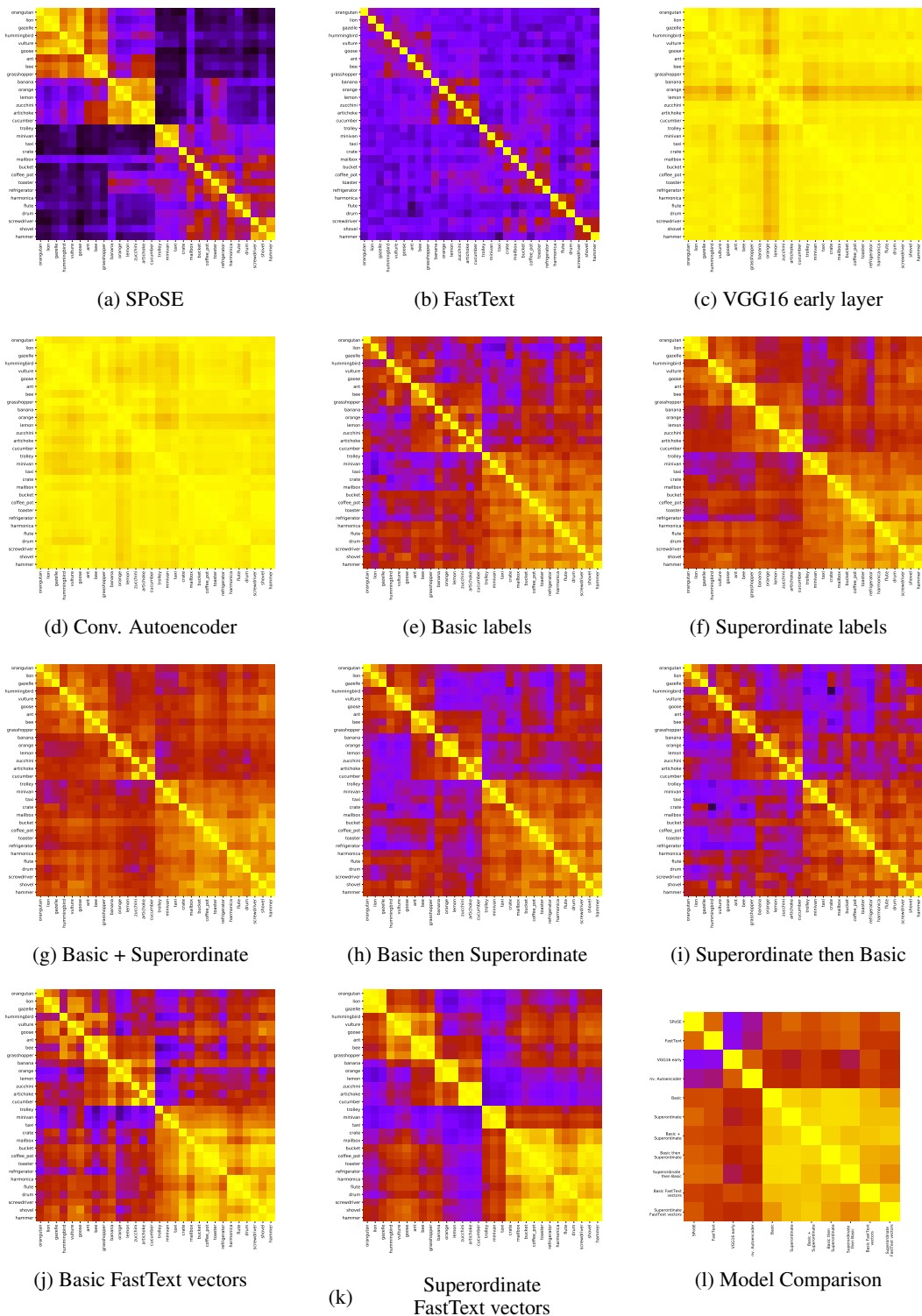

(a) SPoSE

(b) FastText

(c) VGG16 early layer

(d) Conv. Autoencoder

(e) Basic labels

(f) Superordinate labels

(g) Basic + Superordinate

(h) Basic then Superordinate

(i) Superordinate then Basic

(j) Basic FastText vectors

(k) Superordinate FastText vectors

(l) Model Comparison

Figure 2: **Cosine similarity matrix visualizing relationships between between 30 basic level categories.** Lighter yellow colors denote higher similarity, and darker purple colors denote lower similarity. Categories from the same superordinate class are located near to each other in xticks with the order of 'mammal', 'bird', 'insect', 'fruit', 'vegetable', 'vehicle', 'container', 'kitchen appliance', 'musical instrument', and 'tool'.

they are prone to choose 'lemon' because it is the most odd one in superordinate-level. In fact, when the number of superordinate categories in a triplet is two as in the example above, 90% of human responses can be predicted just by telling which one is the odd superordinate category. To investigate how much this task setting would affect the results, we broke down the triplet data based on the number of superordinate categories that a triplet belongs to and reported prediction performance for each split, as shown in the Figure 3. Interestingly, the model trained with superordinate labels alone still performed the best (63%) when superordinate-level information was not very helpful, where all three images in a triplet come from three different superordinate categories, e.g, ('mammal','fruit','vehicle'). Moreover, the superordinate labels CNN (59%) outperformed the basic labels CNN (56%) even when the images were to be compared at the basic level, where all three images in a triplet come from the same superordinate categories, e.g., ('lemon','orange','banana'). This implies humans leverage the guidance from coarser superordinate labels in shaping categorical visual representation in both basic and superordinate levels

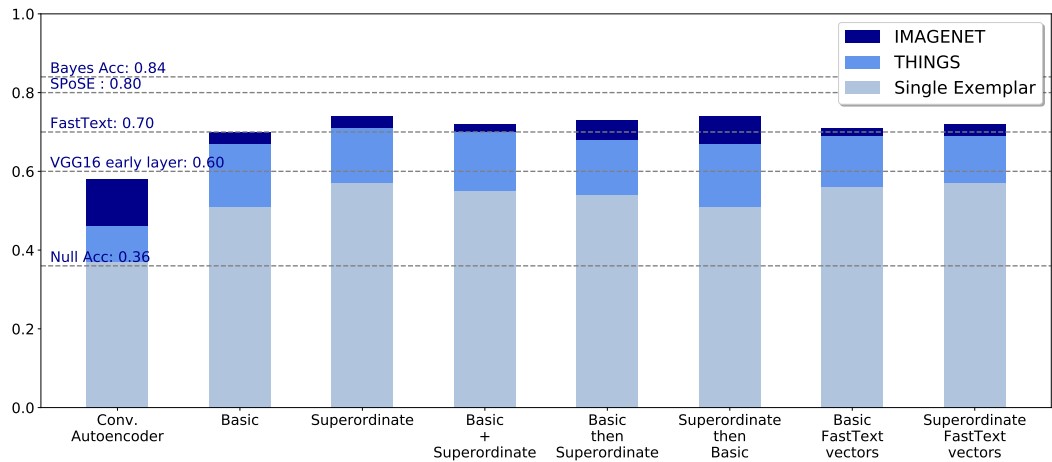

Figure 3: **Comparison of Triplet Prediction Accuracy**. IMAGENET: when using categorical representations averaged over the IMAGENET training dataset (∼1000 images per category); THINGS: when using categorical representation averaged over the THINGS dataset (∼10 images per category); Single Exemplar: when using categorical representation extracted from the single image used for behavioral data collection. Baseline accuracies are indicated by the dashed lines.

## 6    CONCLUSION

To be able to generalize to unseen exemplars, any vision system has to learn statistical regularities that make members of the same category more similar to one another than members of other categories. But where do these regularities come from? Are they present in the bottom-up (visual) input to the network? Or does learning the regularities require top-down guidance from category labels? If so, what kinds of labels? To investigate this problem, we manipulated the visual representations learned by CNNs by supervising them using different types of labels and then evaluated these models in their ability to predict human similarity judgments. We found that the type of label used during training profoundly affected the visual representations that were learned, suggesting that there is categorical structure that is not present in the visual input and instead requires top-down guidance in the form of category labels. We also found guidance from superordinate labels was often as good or better as guidance from much finer-grained basic-level labels. Models trained only on superordinate class labels such as "musical instrument" and "container" were not only more sensitive to these broader classes than models trained on just basic-level labels, but exposure to just superordinate labels allowed the model to learn within-class structure, distinguishing a harmonica from a flute, and a screwdriver from a hammer. This finding is consistent with the previous work that revealed that training with coarser labels induce more semantically structured visual representations (Peterson et al., 2018). More surprisingly, models supervised using superordinate labels (vehicle, tool, etc.) were best in predicting human performance on a triplet odd-one-out task. CNNs trained with superordinate labels not only outperformed other models when the odd-one-out came from a

Table 3: **Triplet Prediction Accuracy**. Macro Mean: global mean of performance ignoring the sample size for each condition. Sample Mean: average performance weighted by sample size for each condition; The best accuracy for each condition among our trained models is in bold text.

| Model | # of unique superordinate categories in the triplet | | | Macro Mean | Sample Mean |
|---|---|---|---|---|---|
| | 1 | 2 | 3 | | |
| Bayes acc (upper bound) | 0.72 | 0.92 | 0.80 | 0.81 | 0.84 |
| SPoSE | 0.59 | 0.90 | 0.75 | 0.75 | 0.80 |
| FastText | 0.56 | 0.88 | 0.56 | 0.67 | 0.70 |
| VGG16 early layer | 0.44 | 0.80 | 0.46 | 0.57 | 0.60 |
| Null acc (lower bound) | 0.40 | 0.35 | 0.36 | 0.37 | 0.36 |
| Conv. Autoencoder | 0.58 | 0.77 | 0.43 | 0.59 | 0.58 |
| Basic labels | 0.56 | 0.87 | 0.59 | 0.67 | 0.70 |
| Superordinate labels | 0.59 | **0.91** | **0.63** | **0.71** | **0.74** |
| Basic + Superordinate | **0.65** | 0.89 | 0.58 | **0.71** | 0.72 |
| Basic then Superordinate | 0.63 | 0.90 | 0.60 | **0.71** | 0.73 |
| Superordinate then Basic | 0.63 | 0.90 | 0.61 | **0.71** | **0.74** |
| Basic FastText vectors | **0.65** | 0.84 | 0.61 | 0.70 | 0.71 |
| Superordinate FastText vectors | 0.61 | 0.90 | 0.59 | 0.70 | 0.72 |
| # of triplets | 507 | 4108 | 5082 | 9697 | 9697 |

different superordinate category (which is not surprising), but also when all three objects from a triplet came from different superordinate categories (e.g., when choosing between a banana, a bee, and a screwdriver). Our ongoing work into how different types of labels shape visual representations is exploring the effect of labels specific to different languages (e.g., English vs. Mandarin), and how these may translate to differential human and CNN classification performance.

ACKNOWLEDGMENTS

Details regarding research support will be added post-review.

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

# 7 SUPPLEMENTARY MATERIAL

## 7.1 LIST OF 30 CATEGORIES

| Superordinate-level Category | Basic-level Category | Wordnet ID |
| --- | --- | --- |
| Mammal | Orangutan | n02480495 |
| | Gazelle | n02423022 |
| | Lion | n02129165 |
| Insect | Ant | n02219486 |
| | Bee | n02206856 |
| | Grasshopper | n02226429 |
| Bird | Hummingbird | n01833805 |
| | Goose | n01855672 |
| | Vulture | n01616318 |
| Vegetable | Artichoke | n07718747 |
| | Cucumber | n07718472 |
| | Zucchini | n07716358 |
| Fruit | Orange | n07747607 |
| | Lemon | n07749582 |
| | Banana | n07753592 |
| Tool | Hammer | n03481172 |
| | Screwdriver | n04154565 |
| | Shovel | n04208210 |
| Vehicle | Minivan | n03770679 |
| | Trolley | n04335435 |
| | Taxi | n02930766 |
| Musical Instrument | Drum | n03249569 |
| | Flute | n03372029 |
| | Harmonica | n03494278 |
| Kitchen Appliance | Refrigerator | n04070727 |
| | Toaster | n04442312 |
| | Coffee pot | n03063689 |
| Container | Bucket | n02909870 |
| | Mailbox | n03710193 |
| | Crate | n03127925 |

## 7.2 CONV. AUTOENCODER PREDICTIONS

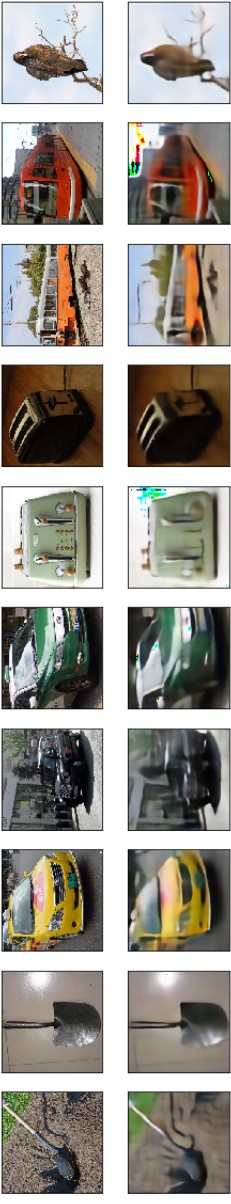

## 7.3 Average precision and average recall scores for the trained models.

The scores were sample-wise averaged (i.e., averaged over samples) for Basic + Superordinate CNN, and macro-averaged (i.e.,averaged over categories) for the other models.

| Model Name | Learning Scheme | # classes | Dimension of Output | Average Precision | Average Recall |
|---|---|---|---|---|---|
| Basic labels | One-step | 30 | 30 | 0.90 | 0.90 |
| Superordinate labels | One-step | 10 | 10 | 0.94 | 0.94 |
| Basic + Superordinate | One-step | 40 | 40 | 0.91 | 0.91 |
| Basic then Superordinate | Two-step | 10 | 10 | **0.95** | **0.95** |
| Superordinate then Basic | Two-step | 30 | 30 | 0.88 | 0.88 |
| Basic FastText vectors | One-step | 30 | 300 | 0.47 | 0.50 |
| Superordinate FastText vectors | One-step | 10 | 300 | 0.72 | 0.75 |

## 7.4 OTHER CLUSTERING QUALITY MEASURES

SC: Silhouette Coefficients; CH: Calinski-Harabasz Index; DB: Davies-Bouldin Index; BW: Between-to-within class dispersion in cosine distance; The arrow indicates in which direction of metric value represent more dense and well separated clusterings.                        NEW

| Model | By superordinate category | | | | By basic category | | | |
|---|---|---|---|---|---|---|---|---|
| | SC↑ | CH↑ | DB↓ | BW↑ | SC↑ | CH↑ | DB↓ | BW↑ |
| Conv. Autoencoder | -0.06 | 166.08 | 12.24 | 0.11 | -0.09 | 70.19 | 15.19 | 0.15 |
| Basic labels | -0.01 | 427.43 | 6.45 | 0.64 | -0.02 | 200.45 | **7.35** | 0.84 |
| Superordinate labels | **0.00** | 628.95 | 5.25 | 0.71 | -0.02 | 226.09 | 11.04 | 0.8 |
| Basic + Superordinate | -0.01 | 534.81 | 5.79 | 0.61 | -0.02 | 231.97 | 7.62 | 0.78 |
| Basic then Superordinate | **0.00** | 580.74 | 5.61 | 0.76 | -0.02 | 233.15 | 8.62 | 0.9 |
| Superordinate then Basic | -0.01 | 525.59 | 5.53 | 0.75 | **-0.01** | 227.35 | 7.47 | 0.93 |
| Basic FastText vectors | -0.01 | 1021.60 | **5.20** | 0.95 | -0.04 | 423.39 | 8.75 | 1.14 |
| Superordinate FastText vectors | -0.01 | **1324.88** | 5.24 | **1.11** | -0.05 | **445.75** | 14.02 | **1.18** |

## 7.5 T-SNE PLOTS FROM OUR TRAINED MODELS

FIX

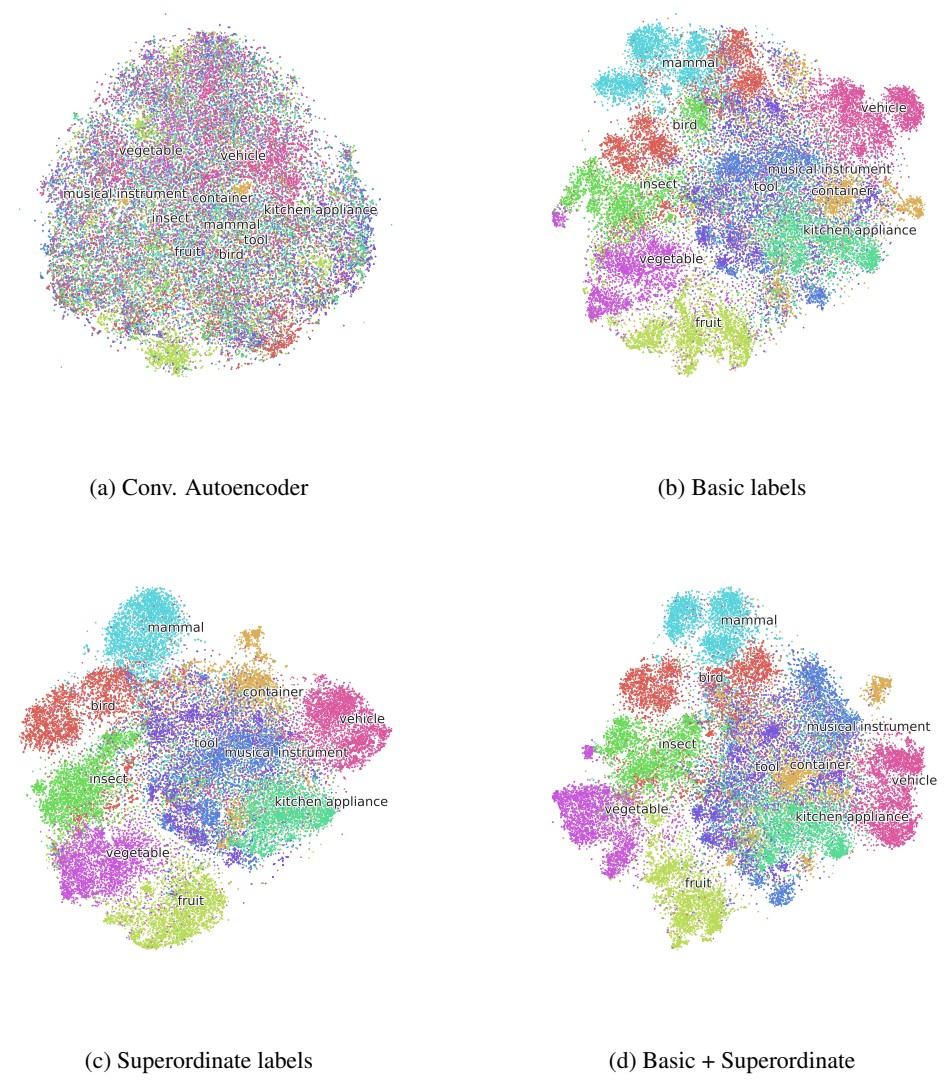

(a) Conv. Autoencoder

(b) Basic labels

(c) Superordinate labels

(d) Basic + Superordinate

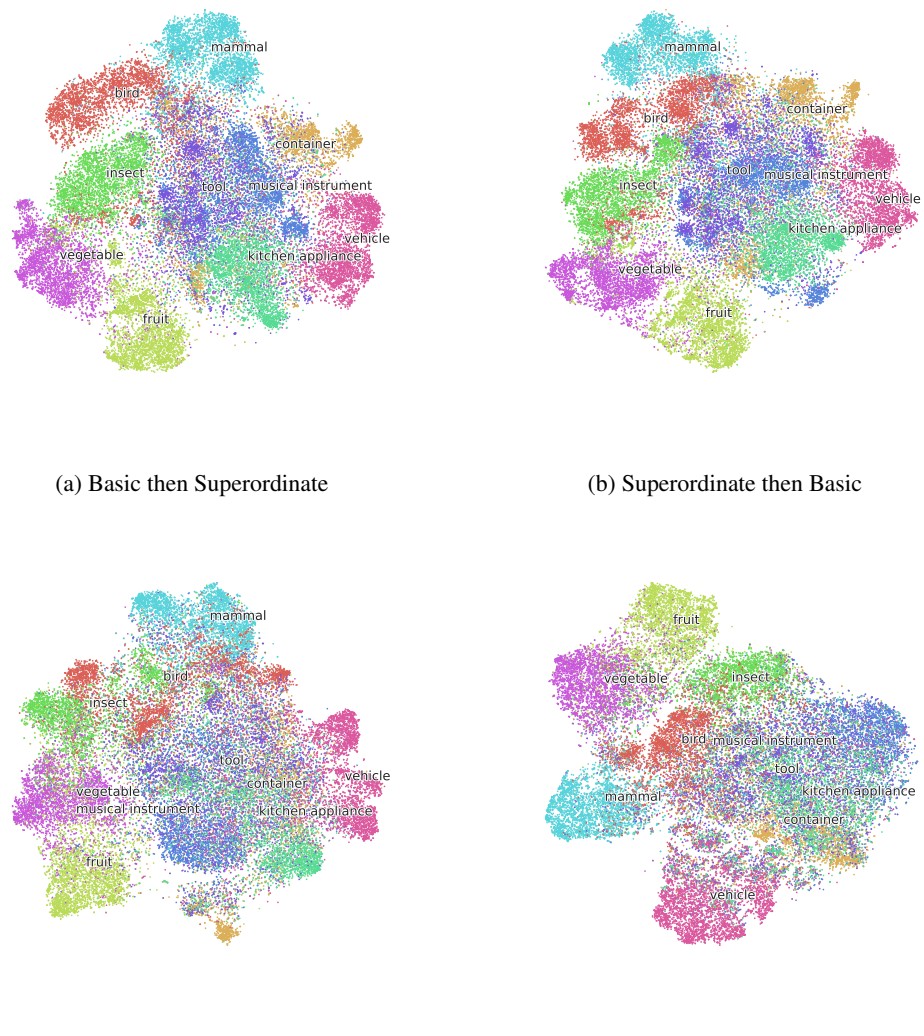

(a) Basic then Superordinate

(b) Superordinate then Basic

(c) Basic FastText vectors

(d) Superordinate FastText vectors

