# OpenReview forum: "Effects of Linguistic Labels on Learned Visual Representations in Convolutional Neural Networks: Labels matter!"
_ICLR.cc/2020/Conference — Reject_

### Official Review · AnonReviewer1 · 2019-10-24
**Official Blind Review #1**

**Rating:** 6

**Review:**

The authors conduct a comparative study of several variants of CNNs trained on imagenent things category with different types of labeling schemes (direct, superordinate, word2vec embedding targets, etc.) They also use a human judgement dataset based on odd-one-out classification for triplets of inputs as comparison to evaluate whether the CNNs are able to capture the linguistic structure in the label categories as determined by the relation of the superordinate labels to the basic labels.

The authors used the t-SNE embeddings to visualize the representations learned and evaluate whether these cluster related classes close enough. Not suprisingly, training with the word2vec targets produced the best representations for similarity between/within category. Interestingly, the autoencoder failed to learn representations that are easily interpretable by the analysis tools they were using.

This is an interesting study. The core claim being made as follows:

"The representations learned by the models are shaped enormously by the kinds of supervision the models get suggesting that much of the categorical structure is not present in the visual input, but requires top-down guidance in the form
of category labels. "

The fact that the representations being learned are shaped strongly by the supervision is probably not surprising or in contention. However, it is not clear that the representations being learned can be exhaustively interpreted by convenient visualization tools. In my opinion, absence of evidence here is not clearly an evidence of absence. However, I still think these are interesting analyses so I am giving weak accept.



**Experience Assessment:**

I do not know much about this area.

**Review Assessment: Checking Correctness Of Derivations And Theory:**

N/A

**Review Assessment: Checking Correctness Of Experiments:**

I assessed the sensibility of the experiments.

**Review Assessment: Thoroughness In Paper Reading:**

I made a quick assessment of this paper.

---

> ### Author Response · Authors · 2019-11-15
> **We thank R1 for providing feedback to improve the work**
>
> We thank R1 for providing feedback to improve the work. Below are our comments answering R1’s main concerns and explaining how we addressed them in the revised paper.
>
> [Addressing the main claim]
> Although R1 found the idea of studying the effect of linguistic labels on visual representation interesting, there was a concern that current analysis might have missed a meaningful structure that the model trained with no label (e.g.,Conv. Autoencoder) have. We agree that this is a possibility and that the results may vary depending on which metric and methods were used.
>
> To address this issue, we visualized and analyzed the visual representations in multiple ways as possible.
>
> 1) In addition to the cosine similarity matrix, we also added a T-SNE plot in the Supplementary 7.5, which was mistakenly missing from the previous paper. Both visualizations showed Conv. Autoencoder visual representations are poorly discriminable not only basic level categories but also superordinate level categories.
> 2) We added in the supplementary 7.4 three more metrics (Silhouette Coefficient, Calinski-Harabasz Index, Davies-Bouldin Index) to analyze the discriminability of the visual representations for each category, in addition to the existing Between-to-within class variance metric. All metrics showed that Conv. Autoencoder has the poorest clustering quality.
>
> Results above together supported that the visual input alone is not sufficient to produce any clusterable structure, not to mention category representations. Of course, it’s still possible that some aspects of the learned visual representations cannot be identified using our current visualization methods, but even if this were the case our main claim would still hold that:
>
> 1) Guidance from superordinate labels was often as good or better as guidance from much finer-grained basic-level labels for shaping semantically structured visual representation.
> 2) Human representations used in an odd-one-out task are highly modulated by semantic information, especially at the superordinate level
>
> [Addressing related work]
> Although R1 did not mention this as a major concern, we made major revisions (we hope for the better!) in describing how our work is situated in the context of existing research. There is now a separate paragraph that reviews related work, and that explains how this work contributes to both computer vision and behavioral science (e.g., psychology, neuroscience). The studies covered in the current related work section include:
>
> 1) studies where semantic label embeddings are used for better image classification
> 2) studies on understanding and uncovering human categorical visual representation

---

### Official Review · AnonReviewer2 · 2019-10-24
**Official Blind Review #2**

**Rating:** 6

**Review:**

This paper assesses the effects of training an image classifier with different label types: 1-hot coarse-grained labels (10 classes), 1-hot fine grained labels (30 labels which are all subcategories of the 10 coarse-grained categories), word vector representations of the 30 fine-grained labels. They also compare the representations learned from an unsupervised auto-encoder. They assess the different representations through cosine similarity within/between categories and through comparison with human judgments in an odd-one-out task. They find that (i) the auto-encoder representation does not capture the semantic information learned by the supervised representations and (ii) representations learned by the model depend on the label taxonomy,  how the targets are represented (1-hot vs. wordvec), and how the model is trained (e.g. fine-grained then coarse grained stages), (iiii) the different representations predict human judgements to differing degrees. the first finding is obvious and I'm not even sure why it needs to be stated -- of course semantics of images are not inherently encoded in the pixels of an image! The second point again, is not surprising . This paper starts to get at some interesting questions but does not follow through.  It is also quite confusing to read despite thee simple subject matter. This paper is also missing a related work section! There has been so much word on adding structure to the label space of image classifiers (e.g. models that learn image/text embedding space jointly, models that predict word vectors, graphical model approaches to building in semantic information, etc.) and none of this is discussed. There has also been work on comparing convnet representations to human percepts e.g. https://cocosci.princeton.edu/papers/Peterson_et_al-2018-Cognitive_Science.pdf)and none of this work is discussed! This work needs to be better situated within the context of previous work in this field. Please write a related work section.

Detailed comments/questions:
- It would be good to add a super-basic model to table 1 for comparison (i.e. first train of coarse level categories and then fine-tine on the more fine-grained taxonomy).
- It would be good to compare the use of word vector representations at both the basic and superordinate levels; the 1-hot vs word vector targets and the basic vs superordinate taxonomy seem like orthogonal axess to explore and I'm not sure why the authors didn't test all combinations.
- the authors found the imagenet categorical representations were most predictive of human judgements in the odd-one-out task. This seems highly unsurprising since (i) the humans saw images from the Imagenet dataset (not THINGS) and (ii) humans leverage semantic information when making similarity judgements.
- What categories had the least inter-rater agreement.. was there any relationship between these categories and the similarity of representations learned by the convnet?
- It seems the odd-one-out comparison always involves averaging image representations at the basic category level. In the case where the items come from three different superordinate classes it would bee interesting to see the results when averaging over superordinate classes as well.
- In table 3, what does the FastText column just list "true"/"false" rather than accuracies? I would expect this column to show the accuracy when the FastText embeddings for the three words are used to compute similarity. I don't understand what the "true"/"false" is meant to indicate. Also it's not clear to be what the two rows in table three are meant to correspond to?
- The authors claim "Surprisingly, the kind of supervised input that proved most effective in matching human performance on the triplet odd-one-out task was training with superordinate labels". This should be qualified to say that, when there are two or more superordinate classes represented in the triplet, the superordinate labels are highly effective when the three items come from three different superordinate classes. I'm also not clear why this would be surprising? Could the authors elaborate?
- I'm surprised more space isn't given to discussing the wordvec representations since these should capture some of the semantic information that the 1-hot encodings might miss. In fact, the word vectors targets seem to perform as good as or close to the other representations on the odd-one-out

In short, I really like the overall idea of comparing convnet representations with human perceptions of images. However, this work barely scratches the surface of what could be done here and mostly reveals incredibly obvious results. There are so many interesting questions to ask regarding the relationship between how humans perceive similarity and what is encoded in a convnet representation. For example, it would have been very interesting to test the effects of asking the human rates to cue in on different aspects of the image. Focusing on semantic similarity, visual similarity, etc. would all likely give different ratings.



----------------------------------------------------------
Update (in light of rebuttal)

I appreciate the authors lengthly and considered response. In particular, the updated related work and expansion of the empirical experiments. While I am more comfortable with this paper being accepted that previously (and have updated by score to "weak accept" to reflect this), I still think the paper has a lot of room for improvement. In particular, I suggest a more expansive analysis of human perceptions and a discussion f the implications of the findings.

**Experience Assessment:**

I have published in this field for several years.

**Review Assessment: Checking Correctness Of Derivations And Theory:**

N/A

**Review Assessment: Checking Correctness Of Experiments:**

I assessed the sensibility of the experiments.

**Review Assessment: Thoroughness In Paper Reading:**

I read the paper at least twice and used my best judgement in assessing the paper.

---

> ### Author Response · Authors · 2019-11-15
> **We thank R2 for pointing out the paper's weaknesses and providing interesting questions/suggestions**
>
> We thank R2 for pointing out the paper's weaknesses and providing interesting questions/suggestions that would be very helpful for exploring our idea. Below are our comments answering R2’s main concerns and explaining how we addressed them in the revised paper. We also provided answers to other questions of R2 in detail below.
>
> [Addressing the main claim]
> One of R2's main concerns is that despite the idea of studying the effect of linguistic labels on visual representation being interesting, most of the findings were obvious and not surprising. We understand our work has limitations, but we also think we were not effective enough at explaining what we found in the current dataset and analysis in the previous draft. In order to address this concern,  we made a serious revision on the introduction and results to clearly convey our main findings. We also polished the formatting of figures and tables to increase the readability. Below is a summary of the findings for our main claim:
>
> 1) We found that the type of label used during training profoundly affected the visual representations that were learned, suggesting that there is a categorical structure that is not present in the visual input and instead requires top-down guidance in the form of category labels.
>
> 2) We also found guidance from superordinate labels was often as good or better as guidance from much finer-grained basic-level labels. Models trained only on superordinate class labels such as "musical instrument" and "container" were not only more sensitive to these broader classes than models trained on just basic-level labels, but exposure to just superordinate labels allowed the model to learn within-class structure, distinguishing a harmonica from a flute, and a screwdriver from a hammer.
>
> 3) More surprisingly, models supervised using superordinate labels (vehicle, tool, etc.) were best in predicting human performance on triplet odd-one-out task. CNNs trained with superordinate labels not only outperformed other models when the odd-one-out came from a different superordinate category (which is not surprising), but also when all three objects from a triplet came from different superordinate categories (e.g., when choosing between a banana, a bee, and a screwdriver).
>
> [Addressing related work]
> As suggested by R2 and other reviewers, we have made serious efforts to revising the introduction and adding a related work paragraph to better explain why this research is meaningful in the context of existing research. The studies covered in the current related work paragraph include:
>
> 1) studies where semantic label embeddings are used for better image classification
> 2) studies on understanding and uncovering human categorical visual representation, which includes more discussions about the paper R2 mentioned (Peterson et al., 2018).

---

> > ### Author Response · Authors · 2019-11-15
> > **[Addressing Questions]**
> >
> > 1) R2: “the authors found the imagenet categorical representations were most predictive of human judgments in the odd-one-out task. This seems highly unsurprising since (i) the humans saw images from the Imagenet dataset (not THINGS) and (ii) humans leverage semantic information when making similarity judgments.”
> >
> > Answer: We think this is an interesting finding because triplet prediction accuracy was highest when models used IMAGENET category representations and lowest when single exemplar representations were used, even if that exemplar image is the one that the participant actually saw during the experiment. This shows that when humans do visual similarity ratings, they not only evaluate visual inputs and use rich and abstract semantic information learned from viewing myriad exemplars.
> >
> > 2) R2: “In table 3, what does the FastText column just list "true"/"false" rather than accuracies? I would expect this column to show the accuracy when the FastText embeddings for the three words are used to compute similarity. I don't understand what the "true"/"false" is meant to indicate. Also, it's not clear to be what the two rows in table three are meant to correspond to?”
> >
> > Answer: In the previous draft, we split the data by each condition first (‘NSUPER’ column; by the number of superordinate labels that three items belong to) and then we split the data again by Fasttext prediction (‘FastText Correct’ column, whether FastText embeddings correctly predict human judgement or not). The reason for this split was to compare the data that can be explained or cannot be explained by semantic information alone (FastText). However we understand this split is arbitrary and confusing, thus we decided to remove this split in the revised paper (Table 3).
> >
> > 3) R2: “the authors claim "Surprisingly, the kind of supervised input that proved most effective in matching human performance on the triplet odd-one-out task was training with superordinate labels". This should be qualified to say that, when there are two or more superordinate classes represented in the triplet, the superordinate labels are highly effective when the three items come from three different superordinate classes. I'm also not clear why this would be surprising? Could the authors elaborate?”
> >
> > Answer: On triplet trials in which two of the images are from the same superordinate category and one is from a different category (the column labeled `2` in Table 3), people overwhelmingly (~90%)  chose the image from the different superordinate class as the odd-one-out. It is, of course, unsurprising that the models trained with superordinate labels perform so well on this task given that all the model needs to ‘know’ is which of the three is from a different class. What *is* surprising, we think, is that training with superordinate labels is so effective when the choice cannot be decided by appeal to the superordinate class (i.e., when all three images are from the same superordinate class (column `1`) or all three are from different superordinate classes (column `3`). For example, when items are from three different superordinate categories like ‘orangutan’, ‘orange’, ‘minivan’, one may need to use other information (either visual or more higher-level semantic information ‘natural’ vs ‘man-made’) to find the odd one. Interestingly, we can observe that our model trained with superordinate one-hot vector predicts human behavior better than other models even in these conditions. This is especially surprising, given that superordinate label one-hot vectors only give coarse-grained supervision (dim =10) compared to other basic label vectors (dim=30), combined vector (dim=40) or wordvectors (dim =300)
> >
> > 4) R2: “I'm surprised more space isn't given to discussing the wordvec representations since these should capture some of the semantic information that the 1-hot encodings might miss. In fact, the word vectors targets seem to perform as good as or close to the other representations on the odd-one-out”
> >
> > Answer: We agree with your observation and in fact we did expect that the models trained with distributed word vectors as targets would perform better than one-hot vector models for the reason the reviewer mentions. However, it turns out the performance of wordvec model was similar or lower than superordinate, which means, in turn, superordinate model was often as good or better as the model with word vectors. As described above, we think this is surprising, given that superordinate label one-hot vectors only give coarse-grained supervision (dim =10) compared to other basic label vectors (dim=30), combined vector (dim=40) or wordvectors (dim =300)

---

> > ### Author Response · Authors · 2019-11-15
> > **[Addressing suggestions]**
> >
> > 1) Add other models in model comparison (e.g, a super-basic model, superordinate wordvec model)
> > Answer: We included them in the analysis in the revised paper, but these results did not change our main claim.
> >
> > 2) Analyze the level of inter-rater agreement in each category and see if this relates to the results from our models
> > Answer: This is an interesting suggestion, but we could not include these results here due to the time constraint.
> >
> > 3) Use representation averaged over superordinate classes to predict performance on the triplet task (especially when items are from three different superordinate classes).
> > Answer: This is an excellent suggestion, but we could not include this analysis at this time due to time constraints, but are working on including it in future work and it should be ready in time for the conference.
> >
> > 4) Include various concepts of similarity to cue in human similarity judgment (semantic similarity, visual similarity).
> > Answer: Although we could not collect more data on the suggested design due to the time constraint, this is an interesting suggestion. Thank you!
> >
> > We thank R2 again for reviews and constructive suggestions and please let us know if we have addressed your concerns, and if there’s any further concerns or questions.

---

### Official Review · AnonReviewer5 · 2019-10-28
**Official Blind Review #5**

**Rating:** 6

**Review:**

Summary: This paper demonstrates the importance of labels at various levels (no label, basic level label, and superordinate level) as well as in combination to determine the importance of semantic information in classification problems. They train an identical CNN architecture either as an autoencoder (no labels), with the basic label, with the subordinate label, with the basic and subordinate labels, and with basic labels which are fine-tuned with one-hot encodings of superordinate labels, as well as with word vectors. Classification accuracy, t-SNE, cosine similarity matrices and predictions on a human behavior task are used to evaluate the differences across labels types. The authors find that superordinate labels are helpful and important for classification problems.

Major comments:
- Authors need to include more related work and describe the main related paper they mention (Peterson et al 2018) as well as describe how their work fits in with previous work
- While the idea here is novel and impactful, the experiments used to explain the importance of superordinate labels do have not much compelling information and are not well described
- 4.2 plots for visualization are mentioned to be in the appendix, but are not there

Minor comments:
-	Fig2 large subordinate group text would help
-	Lots of typos throughout and grammar mistakes
o	Typo ‘use VGG16’ and then ‘Vgg16’ in same paragraph bottom of page 4
o	Typo top of page 2 “Convolutional neural network(CNN)”
o	Appendix list – ‘banna’ typo under Fruit
o	Page 1 intro ‘for both behavioral and computer vision’ doesn’t really make sense
o	Page 3 top section ‘new one’ should be ‘new ones’
o	Bottom of page 3 ‘room from improvement’
o	Last line of conclusion – ‘classificacation’

Consensus: This is a very interesting and potentially impactful idea, but the experiments used to defend and explain the importance of superordinate labels are relatively weak. Significant work on writing and experimental side should be complete, but because this is novel and important work for classification, with some serious revisions, I would suggest accepting this paper.


**Experience Assessment:**

I have published one or two papers in this area.

**Review Assessment: Checking Correctness Of Derivations And Theory:**

N/A

**Review Assessment: Checking Correctness Of Experiments:**

I assessed the sensibility of the experiments.

**Review Assessment: Thoroughness In Paper Reading:**

I read the paper at least twice and used my best judgement in assessing the paper.

---

> ### Author Response · Authors · 2019-11-15
> **We thank R5 for identifying the paper's strengths and weaknesses**
>
> We thank R5 for identifying the paper's strengths and weaknesses and providing suggestions for improvement. Below are replied to R5’s concerns, and a summary of how we addressed them in the revised paper.
>
> [Addressing related work]
> As suggested by R5 and other reviewers, we have made serious efforts to revising the introduction and adding a related work paragraph to better explain why this research is meaningful in the context of existing research. The studies covered in the current related work paragraph include:
>
> 1) studies where semantic label embeddings are used for better image classification
> 2) studies on understanding and uncovering human categorical visual representation
>
> We think these changes answer the question asked by R5 in a minor comment, about how our study can benefit both computer vision and behavioral science (e.g., psychology, neuroscience). Our study broadly benefits both computer vision and behavioral science (e.g., psychology, neuroscience) by suggesting that the semantic structure of labels and datasets should be carefully constructed if the goal is to build vision models that learn visual features representations having the potential for human-like generalization. For behavioral science, this research provides a useful computational framework for understanding the effect of training labels on the human learning of category relationships in the context of thousands of naturalistic images of objects.
>
> [Addressing the main claim]
> We thank R5 for pointing out that the evidence supporting our main claim is relatively weak to strongly support our main claim that superordinate labels play a key role in shaping human-like visual representation. Our work has limitations in that our findings are limited to current methods and data, but we also think we were not effective enough at explaining what we found in the current dataset and analysis in the previous draft. In order to address this concern,  we made a serious revision on the introduction and results to clearly convey our main findings. We also polished the formatting of figures and tables to increase the readability. Below is a summary of the findings for our main claim:
>
> 1) We found that the type of label used during training profoundly affected the visual representations that were learned, suggesting that there is a categorical structure that is not present in the visual input and instead requires top-down guidance in the form of category labels.
>
> 2) We also found guidance from superordinate labels was often as good or better as guidance from much finer-grained basic-level labels. Models trained only on superordinate class labels such as "musical instrument" and "container" were not only more sensitive to these broader classes than models trained on just basic-level labels, but exposure to just superordinate labels allowed the model to learn within-class structure, distinguishing a harmonica from a flute, and a screwdriver from a hammer.
>
> 3) More surprisingly, models supervised using superordinate labels (vehicle, tool, etc.) were best in predicting human performance on triplet odd-one-out task. CNNs trained with superordinate labels not only outperformed other models when the odd-one-out came from a different superordinate category (which is not surprising), but also when all three objects from a triplet came from different superordinate categories (e.g., when choosing between a banana, a bee, and a screwdriver).
>
> [Others]
> We filled in the missing figure of Supplementary 7.5 that R5 mentioned as the last major concern. We also polished the paper to fix all the spelling and grammatical errors that R5 pointed out as minor points. Thank you for pointing these out.
>
> We thank R5 again for the review and please let us know if we have addressed your concerns, and if there’s any further concerns or questions.

---

### Author Response · Authors · 2019-11-15
**General Comments**

We thank the reviewers for their time in reading our work and providing feedback. We first provide general comments on the reviewers’ major concerns and the way we addressed them, followed by detailed replies to each reviewer's individual comments and questions. We also mark the major addition and fix in the margin of the paper as “NEW” and “FIX”

All reviewers agreed that the central claim of this paper was interesting: the superordinate-level structure of labels plays a key role in shaping human-like visual representations. However, the reviewers also pointed out that: 1) our paper could be improved by better situating our study in the context of existing research (R2, R5), and 2) evidence for the main claim was relatively weak and was not exhaustively explored (R1, R2, R5). To address these weaknesses, we made major revisions to the paper:

[Addressing related work]
We now have a separate ‘related work’ paragraph as part of the introduction, which we hope better explains the relationship between our work and work in computer vision and the behavioral sciences (e.g., psychology, cognitive neuroscience). The studies covered in the current related work section include:

1) studies where semantic label embeddings are used for better image classification
2) studies on understanding and uncovering human categorical visual representation

[Addressing the main claim]
We revised the introduction and results to more clearly convey the logic underlying the design of our study and the interpretation of the results. We also polished the formatting of figures and tables to increase their readability. We also added additional analyses in the supplementary material further supporting our main claims. Below is a summary of what our main finding is and how it is supported.

1) We found that the type of label used during training profoundly affected the visual representations that were learned, suggesting that there is a categorical structure that is not present in the visual input and instead requires top-down guidance in the form of category labels.

2) We also found guidance from superordinate labels was often as good or better as guidance from much finer-grained basic-level labels. Models trained only on superordinate class labels such as "musical instrument" and "container" were not only more sensitive to these broader classes than models trained on just basic-level labels, but exposure to just superordinate labels allowed the model to learn within-class structure, distinguishing a harmonica from a flute, and a screwdriver from a hammer.

3) More surprisingly, models supervised using superordinate labels (vehicle, tool, etc.) were best in predicting human performance on triplet odd-one-out task. CNNs trained with superordinate labels not only outperformed other models when the odd-one-out came from a different superordinate category (which is not surprising), but also when all three objects from a triplet came from different superordinate categories (e.g., when choosing between a banana, a bee, and a screwdriver).

---

### Decision · Program_Chairs · 2019-12-19

**Decision:**

Reject

**Comment:**

This paper explores training CNNs with labels of differing granularity, and finds that the types of information learned by the method depends intimately on the structure of the labels provided.

Thought the reviewers found value in the paper, they felt there were some issues with clarity, and didn't think the analyses were as thorough as they could be. I thank the authors for making changes to their paper in light of the reviews, and hope that they feel their paper is stronger because of the review process.